

# The Eurasian Modern Pollen Database (EMPD), Version 2

Basil A.S. Davis[1], Manuel Chevalier[1], Philipp Sommer[1], Vachel A Carter[2], Walter Finsinger[3], Achille Mauri[4], Leanne N. Phelps[1], Marco Zanon[5], Roman Abegglen[6], Christine M. Åkesson[7], Francisca Alba-Sánchez[8], R. Scott Anderson[9], Tatiana G. Antipina[10], Juliana R. Atanassova[11], Ruth Beer[6], Nina I. Belyanina[12], Tatiana A. Blyakharchuk[13], Olga K. Borisova[14], Elissaveta Bozilova[15], Galina Bukreeva[16], M. Jane Bunting[17], Eleonora Clò[18], Daniele Colombaroli[19], Nathalie Combourieu-Nebout[20], Stéphanie Desprat[21], Federico Di Rita[22], Morteza Djamali[23], Kevin J. Edwards[24], Patricia L. Fall[25], Angelica Feurdean[26], William Fletcher[27], Assunta Florenzano[18], Giulia Furlanetto[28], Emna Gaceur[29], Arsenii T. Galimov[10], Mariusz Gałka[30], Iria García-Moreiras[31], Thomas Giesecke[32], Roxana Grindean[33], Maria A. Guido[34], Irina G. Gvozdeva[35], Ulrike Herzschuh[36], Kari L. Hjelle[37], Sergey Ivanov[38], Susanne Jahns[39], Vlasta Jankovska[40], Gonzalo Jiménez-Moreno[41], Monika Karpińska-Kołaczek[42], Ikuko Kitaba[43], Piotr Kołaczek[42], Elena G. Lapteva[44], Małgorzata Latałowa[45], Vincent Lebreton[46], Suzanne Leroy[47], Michelle Leydet[48], Darya A. Lopatina[49], José Antonio López-Sáez[50], André F. Lotter[6], Donatella Magri[22], Elena Marinova[51], Isabelle Matthias[52], Anastasia Mavridou[53], Anna Maria Mercuri[18], Jose Manuel Mesa-Fernández[41], Yuri A. Mikishin[35], Krystyna Milecka[42], Carlo Montanari[54], César Morales-Molino[6], Almut Mrotzek[55], Castor Muñoz Sobrino[31], Olga D. Naidina[56], Takeshi Nakagawa[43], Anne Birgitte Nielsen[57], Elena Y. Novenko[58], Sampson Panajiotidis[53], Nata K. Panova[10], Maria Papadopoulou[53], Heather S. Pardoe[59], Anna Pędziszewska[45], Tatiana I. Petrenko[35], María J. Ramos-Román[60], Cesare Ravazzi[28], Manfred Rösch[61], Natalia Ryabogina[38], Silvia Sabariego Ruiz[62], J. Sakari Salonen[60], Tatyana V. Sapelko[63], James E. Schofield[24], Heikki Seppä[60], Lyudmila Shumilovskikh[64], Normunds Stivrins[65], Philipp Stojakowits[66], Helena Svobodova Svitavska[67], Joanna Święta-Musznicka[45], Ioan Tantau[33], Willy Tinner[6], Kazimierz Tobolski[69], Spassimir Tonkov[15], Margarita Tsakiridou[53], Verushka Valsecchi[6], Oksana G. Zanina[68], Marcelina Zimny[45]

[1]University of Lausanne, Institute of Earth Surface Dynamics IDYST, Faculté des Géosciences et l'Environnement, Batiment Géopolis, CH-1015, Lausanne, Switzerland
[2]Charles University, Department of Botany, Benatska 2, Praha 2 CZ128-01, Czech Republic
[3]ISEM, CNRS, University of Montpellier, EPHE, IRD, Montpellier, France
[4]European Commission Joint Research Centre, Directorate D – Sustainable Resources - Bio-Economy Unit, Via E. Fermi 2749, I-21027 Ispra (VA), Italy, Italy
[5]Kiel University, Institute of Pre- and Protohistoric Archaeology, Johanna-Mestorf-Str. 2-6, 24118, Kiel, Germany
[6]University of Bern, Institute of Plant Sciences, Altenbergrain 21, Bern, Switzerland
[7]University of St Andrews, North Street, St Andrews, KY16 9AL, UK
[8]University of Granada, Department of Botany, Avda. Fuente Nueva, 18071-Granada, Spain
[9]School of Earth and Sustainability, 624 S. Knoles St., Ashust Building, Room A108, Flagstaff, Arizona, USA
[10]Botanical Garden of the Ural Branch of the Russian Academy of Sciences, 620144, Ekaterinburg, Russia, 8-Marta str., 202a, Russian Federation
[11]University of Sofia, Biological Faculty, Department of Botany, 8 Dragan Tzankov bld., 1164 Sofia, Bulgaria
[12]Pacific Institute of Geography FEB RAS, 7, Radio Street, Vladivostok 690042, Russia
[13]Instititute of monitoring of climatic and ecological systems of Siberian branch of Russian academy of sciences., Akademicheski ave. 10/3 Tomsk 634055 , Russia




[14]Russian Academy of Sciences, Institute of Geography, Staromonetny lane 29, 119017 Moscow, Russia

[15]Sofia University, Faculty of Biology,  Laboratory of Palynology, 8 Dragan Tsankov blvd.,1164-Sofia, Bulgaria, Bulgaria

[16]Siberian Branch of the Russian Academy of Sciences, c/o N. Ryabogina, Tyumen Scientific Centre SB RAS, Malygina st. 86, 625026 Tyumen, Russia

[17]University of Hull , Department of Geography, Geology and Encironment, Cottongham Road, Hull, HU67RX, UK

[18]Università di Modena e Reggio Emilia, Laboratorio di Palinologia e Paleobotanica – Dipartimento Scienze della Vita, via Campi 287, 41125 Modena, Italy

[19]Royal Holloway University of London, Department of Geography, Egham, Surrey TW20 0EX, UK

20UMR 7194 - CNRS/MNHN, Dpt Homme et Environnement, Institut de Paléontologie Humaine 1, rue René Panhard, F-75013 Paris, France

[21]University of Bordeaux, EPOC UMR 5805, EPHE- PSL University, Allée Geoffroy St Hilaire, 33615 Pessac, France

[22]Sapienza University, Department of Environmental BiologyPiazzale Aldo Moro, 5, Rome, Italy, Italy

[23]Institut Méditerranéen de Biodiversité et d'Ecologie, Aix-Marseille Université - Campus Aix Technopôle de l'environnement Arbois Méditerranée Avenue Louis Philibert Bât Villemin - BP 80 F-13545 Aix en Provence cedex 4, France

[24]University of Aberdeen, Departments of Geography and Environment and Archaeology, School of Geosciences, Elphinstone Road, Aberdeen AB24 3UF, United Kingdom

[25]University of North Carolina, Department of Geography & Earth Sciences, Charlotte, USA

[26]Goethe University, Department of Physical Geography, Altenhöferallee 1,  60438, Frankfurt am Main, Germany

[27]University of Manchester, Quaternary Environments and Geoarchaeology Group, Department of Geography, School of Environment, Education and Development, Oxford Road, Manchester, M13 9PL, United Kingdom

[28]CNR-IGAG, Laboratory of Palynology and Palaeoecology, Piazza della Scienza 1, 20126 Milano, Italy

[29]GEOGLOB, Faculty of Sciences of Sfax, Route Soukra, BP. 802, 3038, Tunisia

[30]University of Lodz, Faculty of Biology and Environmental Protection, Department of Geobotany and Plant Ecology , Banacha Str. 12/16, 90-237 Lodz, Poland

[31]Universidade de Vigo, Dpto. Bioloxía Vexetal e Ciencias do Solo, Facultade de Ciencias, E-36310, Vigo, Spain

[32]Utrecht University, Department of Physical Geography, Faculty Geoscience, Utrecht University, P.O.Box 80115, 3508 TC, Utrecht , Netherlands

[33]Babes-Bolyai University, Department of Geology1, Kogalniceanu Street, 400084, Cluj-Napoca, Romania

[34]CIR-LASA - University of Genova, Via Balbi, 6, 16126, Genova, Italy, Italy

[35]Far East Geological Institute FEB RAS, 159, Prospekt 100-letiya, Vladivostok 690022, Russia

[36]Alfred Wegener Institute Helmholtz Centre for Polar and Marine Research, Telegraphenberg A45, Germany

[37]University of Bergen, Department of Natural History, University Museum, P.O.Box 7800, 5020 Bergen, Norway

[38]Tyumen Scientific Centre SB RAS, Malygina st. 86, 625026 Tyumen, Russia

[39]Brandenburgisches Landesamt für Denkmalpflege, Wünsdorfer Platz 4-5, 15806 Zossen OT Wünsdorf, Germany

[40]Academy of the Sciences of the Czech Republic, Paleoecological Laboratory, Institute of Botany, Lidická 25/27, 602 00 BRNO, Czech Republic

[41]Universidad de Granada, Departamento de Estratigrafía y Paleontología, Avda. Fuentenueva S/N, 18002, Spain

[42]Adam Mickiewicz University, Laboratory of Wetland Ecology and Monitoring, B. Krygowskiego 10/247, 61-680 Poznań, Poland

[43]Ritsumeikan Univerisity, Research Centre for Palaeoclimatology, 1-1-1 Noji-Higashi, Kusatsu, Shiga 525-8577, Japan

[44]Institute of Plant and Animal Ecology of the Ural Branch of the Russian Academy of Sciences, Laboratory of Paleoecology, 8 Matra str., 202, 620144, Ekaterinburg, Russia

[45]University of Gdańsk, Department of Plant Ecology, Laboratory of Palaeoecology & Archaeobotanyul, Wita Stwosza 59, 80-308 Gdańsk, Poland

[46]CNRS/Muséum National d'Histoire Naturelle, UMR 7194 - Institut de Paléontologie Humaine 1, rue René Panhard, F-75013 Paris, France

[47]AMU-LAMPEA, Aix Marseille Univ, CNRS, Minist Culture, LAMPEA, UMR 7269, 5 rue du Château de l'Horloge, 13094, Aix-en-Provence, France

[48]Aix Marseille Univ, Avignon Université, CNRS, IRD, IMBE, Europôle Arbois, Aix-en-Provence, France



[49]Laboratory of Stratigraphy and Paleogeography of oceans Geological Institute Russian Academy of Sciences , Pyzevskii per., Moscow, 119017 , Russia

[50]Instituto de Historia-CSIC, Albasanz 26-28, 28037 Madrid, Spain

[51]State Office for Cultural Heritage Baden Württemberg, Laboratory for Archaeobotany, Fischersteig 9, 78343 Hemmenhofen, Germany

[52]University of Göttingen, Department of Physical Geography, Institute of Geography, University of Göttingen, Göttingern, Germany

[53]Aristotle University of Thessaloniki, Laboratory of Forest Botany-Geobotany, Faculty of Forestry and Natural Environment, 100 Thessaloniki, Greece

[54]University of Genova, DISTAV - Corso Europa, 26, Italy

[55]University of Greifswald, Institute of Botany and Landscape Ecology, Soldmannstr. 15, 17487 Greifswald, Germany

[56]Geological Institute RAS, Pyzhevsky 7, 119017,Moscow, Russia

[57]Lund University, Sölvegatan 12, 66362 Lund, Sweden

[58]Lomonosov Moscow State University, Faculty of geography, Department of Physical Geography and Landscape Science, Leninskiye gory, 1., 119991, Moscow, Russia

[59]National Museum Wales, Cathays Park, Cardiff CF10 3NP, UK

[60]University of Helsinki, Department of Geosciences and Geography, P.O. Box 64 (Gustaf Hällströmin katu 2), FI-00014, Helsinki, Finland

[61]Universität Heidelberg, Sandgasse 7, 69117 Heidelberg, Germany

[62]Universidad Complutense de Madrid, Dep. de Biodiversidad, Ecología y Evolución. Ciudad Universitaria 28040 - MADRID, Spain

[63]Institute of Limnology, RAS, 9,Sevastyanova st., 196105, St.Petersburg, Russia

[64]University of Goettingen, Wilhelm-Weber-Str. 2a, 37073 Goettingen, Germany

[65]University of Latvia, Department of Geography, Jelgavas str. 1, LV-1004, Riga, Latvia

[66]University of Augsburg, Institute of Geography, Alter Postweg 118, 86159 Augsburg, Germany

[67]Czech Academy of Sciences, Institute of Botany, Zámek 1, CZ-252 43 Pruhonice, Czech Republic

[68]RAS, Laboratory of Soil Cryology, Institute of Physico-Chemical and Biological Problems in Soil Science, Moscow region, Pushchino, Institutskaya 2, 142290, Russia

[69]Deceased

*Correspondence to*: Basil A.S. Davis (basil.davis@unil.ch)

**Abstract.** The Eurasian (née European) Modern Pollen Database (EMPD) was established in 2013 to provide a public database of high-quality modern pollen surface samples to help support studies of past climate, land-cover and land-use using fossil 125 pollen. The EMPD is part of, and complementary to, the European Pollen Database (EPD) which contains data on fossil pollen found in Late Quaternary sedimentary archives throughout the Eurasian region. The EPD is in turn part of the rapidly growing Neotoma database, which is now the primary home for global palaeoecological data. This paper describes version 2 of the EMPD in which the number of samples held in the database has been increased by 60% from 4826 to 8134. Much of the improvement in data coverage has come from Northern Asia, and the database has consequently been renamed the Eurasian 130 Modern Pollen Database to reflect this geographical enlargement. The EMPD can be viewed online using a dedicated map-based viewer at https://empd2.github.io, and downloaded in a variety of file formats at https://doi.pangaea.de/10.1594/PANGAEA.909130 (Chevalier et al., 2019).



## 1 Introduction

Modern pollen samples provide an essential source of information for interpreting and understanding the fossil pollen record,
which in turn provides one of the most important spatially resolved sources of information on Quaternary vegetation and
climate. We use the term fossil pollen here as it is used in the Quaternary sciences, although it is in fact more accurately
described as sub-fossil having usually undergone only limited (if any) mineralization, and including many spores as well as
pollen from flowering plants. Fossil pollen can be found preserved in sediments in lakes and bogs and other anaerobic
environments throughout the Eurasian region extending back throughout the Quaternary. Modern pollen is simply the
component of that fossil record found in the last 100-150 years, most often in the surface layers of lake and bog sediments, but
also including comparable collectors of pollen such as moss polsters.

Davis et al. (2013) include a comprehensive introduction to the different scientific uses of modern pollen samples. Modern
pollen samples have been used to interpret past change in land cover, land use and human impact, the impact on vegetation of
past edaphic change, the effects of fire, pests and disease, as well as hydroseral change. Modern samples have also been used
to understand taphonomic problems with regard to pollen transport, deposition and preservation. One of the early motivations
for establishing large modern pollen datasets and one that still remains important is their use as calibration 'training sets' for
the quantitative reconstruction of past climate. This approach has also more recently been adapted to quantitative
reconstructions of land-cover, where a similar modelling approach to climate reconstruction is applied to determine, for
instance, forest cover. Similarly, modern samples have also been used to establish and model the relationship between
vegetation and pollen assemblages based on the different pollen productivity of different taxa, and thereby provide quantitative
estimates of past vegetation composition in a landscape from records of fossil pollen.

Historically, modern pollen data was often gathered directly for a particular research project but the data was rarely shared,
and if published, often in grey literature such as a thesis, report or monograph. Efforts to develop larger datasets at continental
scales were pioneered in the 1990's, primarily by research groups looking to use these datasets as calibration datasets for
quantitative climate reconstruction. Development however was haphazard, and the datasets had a reputation for being poorly
documented and quality controlled, often containing duplicates, digitized data (not original raw counts), uncertain taxonomic
standardisation, poor geolocation information and loose definitions of 'modern' that could embrace as much as the last 500
years. It became increasingly clear that a quality controlled and standardized database of modern pollen samples was required,
comparable to the European Pollen Database (EPD) for fossil pollen samples, and reflecting the same open-access and
community-based principles.

The Eurasian (née European) Modern Pollen Database (EMPD) was therefore established in 2013 as a complement to the
European Pollen Database (EPD) for fossil pollen (Davis et al., 2013). The first version of the EMPD (referenced herein as the

EMPD1) contained almost 5000 samples, submitted by over 40 individuals and research groups from all over Europe. Over the last 6 years more data has continued to be submitted, and additional efforts have been made to incorporate more data held in open data repositories such as PANGAEA, and made available as supplementary information in published studies. This paper documents the first update to the EMPD (referenced herein as EMPD2), in which the number of samples stored in the
database has increased by around 60%.

The EMPD remains the only open access database of modern pollen samples covering the Eurasian continent. Smaller compilations of modern pollen samples exist for some regions, but these generally have limitations in terms of some or all of the following: 1) the extent of metadata provided, 2) the completeness of the taxa assemblage, 3) the standardization of taxa
nomenclature and hierarchy with respect to the EPD, 4) the inclusion of digitized rather than original raw count data, 5) the inclusion of percentages rather than raw counts, 6) information about the original source of the data and the analyst, and in some cases, 6) limitations to public access. Importantly, all of these aspects limit their compatibility with the EPD, and compatibility with the EPD is one of the primary objectives of the EMPD. The EMPD contains only the original raw count data (no percentage data) for the complete pollen assemblage. The EMPD also contains comprehensive and standardized
metadata about the pollen sample location, the landscape and vegetation environment from which it was collected, the way it was collected, the year that it was collected, as well as who collected and analyzed the sample and where it was published.

The EMPD has no formal spatial domain, but in general it covers the same geographic region as the EPD. This has traditionally been the Palearctic vegetation region of Eurasia excluding China, which has established its own semi-private regional database.
As well as the terrestrial Eurasian landmass and associated islands, it also includes marine samples from coastal margins and enclosed seas. Increasingly however these geographical administrative boundaries have become blurred as regional pollen databases become integrated into the global Neotoma Palaeoecology Database (Williams et al., 2018), hereafter referred to as 'Neotoma'. While regional databases such as the EPD will outwardly retain their identity within Neotoma, internally the data will be completely integrated at a global level. It is also planned that the EMPD will become integrated into Neotoma in the
near future, and with this in mind, the EMPD2 also includes data from outside of the traditional EPD region on the basis that it represented the most expeditious route to making this data publicly available within Neotoma. Consequently, this second version of the EMPD includes not only data from Europe and northern Asia, but also data from Greenland, India, China and North Africa.

## 2 Methods

Details about the structure and metadata of the database have already been described in detail by Davis et al. (2013). The list of metadata fields is shown in **Table 1**. We also include climate and vegetation data for each sample location. The climate data includes mean monthly, seasonal and annual temperature and precipitation climatology from WorldClim2 (Fick and Hijmans, 2017). The climate was assigned according to the nearest grid point within the 30 second (approximately 1km$^2$) resolution of

the WorldClim2 grid. The vegetation data includes realm, biome and ecoregion, taken from Olson et al. (2001). Note that all
samples have been assigned a biome, including marine samples. The biome assigned to marine samples was based on the
nearest point of land to the sample. No climate has been assigned to marine samples.

The protocol for the database follows that of the European Pollen Database, with some additions. The EMPD only includes
samples younger than 200 BP, and with a sampling resolution comparable with the fossil pollen in the EPD. For instance, the
EMPD does not include pollen trap data gathered at monthly or annual resolution, but it does accept trap data averaged over a
period of at least 10 years, which is more comparable with the time typically represented in a fossil pollen sample taken from
a sediment core.

Like the EPD, the EMPD only includes raw count data representing the full pollen assemblage, and it does not contain
percentage data or truncated or summary assemblages. Percentages are excluded because their calculation can vary from author
to author, and therefore unlike raw count data it is not always possible to directly compare different samples from different
sources with percentage data. This is an important data quality criteria, but it has led to the exclusion of some large regional
modern pollen datasets that have been recently published. This is discussed in the next section.

Modern pollen samples have been gathered from a variety of depositional environments, and the type of environment is
recorded for 75% of the samples in the database. The most common environments are moss pollsters (31%), soil (21%) and
lake sediments (19%).

## 2.1 Data sources

The pollen data for the latest update of the EMPD has come from a diverse range of sources, but mainly submissions from
individual researchers and research groups. Most of this has been the result of published research (**Table 2**), but we also include
unpublished data. Additional pollen data has come from open access sources such as the PANGAEA data archive and data
supplements to publications, as well as new fossil pollen data submitted to the EPD and Neotoma since EMPD1 where the
sample age of a sediment core top fulfills the requirements of a modern pollen sample.

Some large independent surface sample datasets covering the Eurasian region have been published and made available since
EMPD1, most notably Binney et al. (2017), Marinova et al. (2018) and Herzschuh et al. (2019). Both Binney et al. (2017) and
Marinova et al. (2017) already include a large amount of data from the EPD and EMPD1, but also data that has not been
publicly released before. This includes 'heritage' data from earlier studies such as the Biome6000 project (Prentice and Webb,
1998), and PAIN projects (Bigelow et al., 2003). This heritage data is mostly composed of percentages, at least some (unknown
part) of which has been digitized, and whose origins, selection criteria and context are rarely documented. Another problem
with this heritage data apart from the limited metadata, is that the definition of a 'modern sample' was more loosely defined





in these early projects, being defined in both PAIN and Biome6000 as anything younger than 500 BP. Unfortunately, the age criteria for selecting individual samples was not recorded when the datasets were compiled.

These problems also extend to the recent release of data by Herzschuh et al. (2019) from China and Mongolia. This data represents most of the modern pollen data held in the Chinese Pollen Database (CPD) (Ni et al., 2010; Zheng et al., 2014). The Herzschuh et al. (2019) dataset includes 2559 modern pollen samples, and is of major importance as the first significant amount of publicly available data from this region. However, the data is only provided as percentages based on a summary of the taxa from each sample, and also includes digitized data. We were therefore unable to include it in the EMPD2. The Herzschuh et

al. (2019) data is available from PANGAEA, along with the Tarasov et al., (2011) dataset of 798 samples mainly from Japan and Eastern Russia, which are also provided as percentages for a limited selection of taxa. We hope that the raw count data for the full assemblage will be made available in the near future.

Other regional pollen databases that overlap with the EMPD includes the Indian Pollen Database (IPD) and the African Pollen

Database (APD). The IPD is still under development and is not publicly accessible, but it includes both fossil and modern pollen samples from the Indian subcontinent (Krishnamurthy and Gaillard, 2011). The EMPD also includes samples from North Africa, which overlaps with the APD (Vincens et al., 2007). Fossil pollen data from the APD is available as individual files and as a partially complete paradox database from the APD website (**Table 3**), but the status of the modern pollen data held within the APD (Gajewski et al., 2002) remains somewhat unclear, since this data has not been made publicly available.

At present the APD is being integrated into Neotoma, and it is hoped that once this is completed the modern pollen data from Africa will become more freely available.

### 2.2 Data processing

As with the EMPD1, the data submitted to the EMPD2 has come in a wide variety of data formats, and with varying levels of metadata. All of these files had to be processed and a variety of quality control checks made (see Davis et al., 2013) before

entry into the database. One of the most time-consuming tasks was to ensure standardization of the original taxon names submitted by the author. These all had to be checked for language, typographical errors and other standardization issues, and assigned an internationally accepted taxa name according to the EPD common taxa 'p_vars' table. If the name did not exist in the EPD taxa table it was checked (using plantlist.org) that it was spelled correctly and was not a synonym. It was then checked against the Neotoma pollen taxa table and assigned the Neotoma accepted taxa name if there was a match. If it was not in the

Neotoma taxa table, and it was established to be a genuine taxa name, then it was added to the EMPD taxa table as a new taxon. Note that although the EMPD is designed to be as compatible with the EPD as possible, the EMPD and EPD do not have a common taxa list and the EMPD has many more taxa than appear in the EPD. In the process of updating the EMPD we have harmonized as much as possible the taxa names in the EMPD with those found in the current EPD, including those names previously in the EMPD1 that have since been included in the EPD. When both the EPD and EMPD are included in the



Neotoma database, then all of the taxa will exist in a single standardized taxa table consisting of all of the taxa in all of the databases.

### 2.3 Changes to EMPD1

As well as adding new data, we also undertook a short review of the data in the original EMPD1. A cross-check between the country attributed to a site and the actual country where the site was located revealed that around 20 sites either had the wrong
location or wrong country code. The geo-location data for around 250 samples in Morocco in EMPD1 has now been removed and placed in the information field. These were all highlighted in EMPD1 as having intractable geolocation errors (Davis et al., 2013), and it was felt that by removing the corrupt information from the geolocation field it would discourage their accidental use. In compensation the EMPD2 now includes new high-quality data from Morocco (see next section).

### 3 Results

**3.1 Spatial sampling**

The amount of data in the database has increased by 60 %, and the EMPD2 now holds 8134 samples compared to 4826 samples in the EMPD1. The country that has experienced the largest increase in samples is Russia, which has gained 2274 more samples on top of the 379 samples already in the EMPD1 (**Fig. 1**). Other significant improvements in data coverage have been made in Italy, Norway and Spain, while data is available for the first time from other countries such as Japan, Cyprus and Kyrgyzstan.
The increase in data from Russia reflects a general improvement in data coverage in EMPD2 from Eastern Europe across to Asia (**Fig. 2**), prompting a re-naming of the database from the "European" to the "Eurasian" Modern Pollen Database.

Countries where there are still relatively few or no samples despite being both relatively populous and having an active palynological community, include Belgium, Netherlands, Hungary, Czech Republic and Slovakia. There are also virtually no
samples from the Balkans. Despite the generally excellent coverage over Scandinavia, north-central Sweden remains poorly sampled, a feature that is also reflected in the lack of fossil pollen data from this area in the EPD. Further east, the distribution of samples tends to be best in the more populous regions and those with better transport infrastructure. Notable areas across northern Eurasia where we still lack samples include the steppes of Ukraine and Kazakhstan and the Central Siberian Plateau. Further south, most of China and Mongolia are well covered by the Chinese Pollen Database (now partly released by Herzschuh
et al., 2019), and as mentioned earlier, there are efforts in India to improve data coverage in this region. A more difficult problem is the lack of samples from many of the Central Asian countries including Turkmenistan, Uzbekistan, Tajikistan, Afghanistan and to some extent Pakistan, where access for scientists is currently difficult or hazardous, and where there are few locally trained scientists. The lack of modern pollen data from these regions is also reflected in a lack of fossil pollen studies from these countries.




## 3.2 Altitudinal sampling

The representativeness of the sample coverage in the vertical spatial domain is not easily discernible from a standard two-dimensional map presented in **Fig. 2**. Vertical climate and vegetation gradients are much steeper than horizontal gradients, and
hilly and mountainous terrain typically hold a greater variety of vegetation and climate types than can be shown on a continental scale map. We make a better attempt to show this by plotting the distribution of samples by altitude on a hypsometric curve for the Palearctic study region (**Fig. 3**). This shows that the number of samples generally follows the proportion of land area represented at each elevation, with more samples at lower altitude, but still the presence of samples as the altitude gets higher. Data coverage has improved in particular in the 500-2500m range between EMPD1 and EMPD2. The upper part of the
altitudinal range above 3500m is dominated by the Himalaya and the Tibetan plateau, which is covered by the Chinese Pollen Database (Herzschuh et al., 2019).

## 3.3 Climate and vegetation sampling

The distribution of the EMPD2 samples across the vegetation biomes of the region (from Olson et al., 2001) is shown in **Figure 4**. Biomes that are well sampled within the Palearctic region include most of those that occur in Europe, namely Mediterranean scrub and Temperate forests and the western range of the Boreal forest/Taiga and Tundra. Less well sampled are the Temperate shrub and grasslands and Deserts of the Central Asian steppe, and the eastern range of the Boreal forest/Taiga and Tundra. Again, the Chinese Pollen Database (Herzschuh et al., 2019) covers much of the Montane biomes of the Himalaya and Tien
Shan ranges, the grasslands and deserts of the Gobi area and Mongolia, and Temperate and Tropical forest biomes of East Asia.

While a conventional map such as **Fig. 4a** can show how samples are distributed across different biomes in geographical space, it does not show how well those samples are distributed in climate space. Large areas of Earth may have the same or similar
climate, and the distribution of samples in conventional space does not necessarily equate to how well climate space has been sampled. Climate space is important because pollen-based climate reconstructions depend on the use of modern pollen calibration datasets that fully sample the available climate space associated with any particular vegetation type. **Figure 4b** shows the same information as **Fig. 4a**, but this time in climate space. This indicates that the EMPD2 samples appear better distributed in climate space than geographical space, but that there are fewer samples to represent the more extreme climates
found at the edges of the modern climate space (such as tundra, deserts and xeric scrublands). This is shown more clearly in **Fig. 5b**, where the Euclidean distance is calculated between the climate of each of the pollen samples in EMPD2, and all of the available climate space of the Palearctic region. The darker regions around the edges of the climate space show where in climate space the EMPD2 still lacks representative samples. These poorly represented climates are then shown in physical space in **Fig. 5a**. This indicates poor representation in the North African and Persian deserts which are outside the Palearctic



study region, but also areas within the Palearctic region including the Central Asian steppe and more mountainous areas of the

Central Siberian Plateau and Siberia east of Yakutsk (130°E).

**4. Discussion**

The increase in size of the EMPD in version EMPD2 has greatly improved the coverage of modern pollen samples across

Eurasia in relation to geographical, vegetation and climate space. This will make possible more accurate reconstructions of

past land cover and climate given the commensurate improvements in available climate and vegetation analogues of fossil

pollen samples. The database continues to increase in size through a mixture of newly submitted samples from old studies that

predate EMPD1, and more recent studies that have occurred since EMPD1 was first made available. It is still likely that older

data will continue to be submitted to the database, especially as it becomes better known, but it is unlikely that the database

will continue to grow at the present rate given that much of the available older data is now expected to have been submitted.

However, surface sample work has traditionally been less likely to be published in international journals, often confined to

Masters or PhD theses or other grey literature, and the amount of data in existence may therefore be difficult to estimate.

To help promote access and use of the EMPD, we have created an online data viewer https://empd2.github.io (**Fig. 6**) (Sommer

et al., n.d.). This allows the database to be viewed using an intuitive clickable map that displays the location of each sample,

associated metadata, and a plot of the pollen data itself. It is also possible to download the data associated with a sample, and

to make suggested corrections. Other options allow the user to select subsets of the database to be viewed, for instance

associated with particular individuals, projects or research groups. The EMPD viewer allows access to the database in an

intuitive way without requiring any particular computer expertise. This has been very important in not only allowing the casual

user to view and access the data in the database, but also in allowing the data submitters to view their data as it exists in the

database after it has been processed, providing a further quality control check. The data viewer is open source, and can be

adapted for other uses.

The EMPD data viewer is embedded in a web framework that is based on the version control system Github, where users and

data contributors can transparently submit new data or raise issues with the existing data. These can then be reviewed in an

open discussion with the database managers. This framework allows ongoing development of the EMPD in the future, and the

usage of a free version control system additionally ensures full transparency, stability and maintainability of access to the data,

independent of funding and changing collaborations.


As well as simply adding more samples as they are submitted, we hope that the future development of the EMPD will also be

more targeted. It is clear that although sample coverage is much improved in EMPD2, gaps still exist in the data coverage for

Eurasia that would be useful to fill (**Figures 3-5**). One way to do this is to encourage fieldwork to collect samples from these

data-poor regions. This approach however is expensive, since the reason why many of these areas remain unsampled is
precisely because of their remoteness and the difficulty and expense involved in accessing them. An alternative that has not
been widely exploited is to analyze soil and sediment samples gathered as a result of fieldwork expeditions organized with a
different objective in mind. We hope that by demonstrating the important sampling gaps in the database it will encourage
individuals and research groups to consider fieldwork and data analysis in these underrepresented regions.

## 5 Ethical statement and how to acknowledge the database


Users of the database are expected to follow the guidelines of the EPD. These state that normal ethics apply to co-authorship
of scientific publications. Palaeoecological datasets are labour-intensive and complex, they take many years to generate and
may have additional attributes and metadata not captured in the EMPD/EPD. Users of data stored in the EMPD/EPD should
consider inviting the original data contributor of any resultant publications if that contributor's data are a major portion of the
dataset analysed, or if a data contributor makes a significant contribution to the analysis of the data or to the interpretation of
results. For large-scale studies using many EMPD/EPD records, contacting all contributors or making them co-authors will
not be practical, possible, or reasonable. Under no circumstance should authorship be attributed to data contributors,
individually or collectively, without their explicit consent.


In all cases, any use of EMPD data should include the following or similar text in the acknowledgments: *Pollen data were
extracted from the Eurasian Modern Pollen Database (part of the European Pollen Database) and the work of the data
contributors and the EMPD/EPD community is gratefully acknowledged.* Upon publication, please send to the EMPD/EPD a
copy of the published work or a link to the electronic resource. Your assistance helps document the usage of the database,
which is critical to ensure continued support from funders and contributers.

## 6 Data availability

The EMPD is available at https://doi.pangaea.de/10.1594/PANGAEA.909130 (Chevalier et al., 2019). The data is available as
1) an Excel spreadsheet, 2) a PostgreSQL dump, and 3) a SQLite3 portable database format. The data can also be viewed
online using an interactive map-based viewer at https://empd2.github.io/?branch=master.

## 7 Conclusions

The EMPD remains the only public, quality controlled and standardized database of modern pollen samples for the Eurasian
region. This paper describes a recent update to the EMPD in which the database has increased almost 60% in size, so that it
now contains data on 8663 modern pollen samples. This reflects an expansion in spatial coverage across northern and eastern



Asia, which has prompted a change in the name of the database from the European to the Eurasian Modern Pollen Database. The improvement in spatial coverage has increased the number of vegetation and climate analogues for fossil pollen samples in the region that will directly improve reconstructions of past vegetation and climate. However, areas of poor data coverage still exist, particularly in the more remote regions of central and northern Asia, and the Middle East. Development of a new map-based online data viewer for the database is already helping improve access to, and participation in, the EMPD, as well as quality control. We expect the EMPD to continue to grow in the future, although probably at a slower rate given that most of the previously published 'heritage' data has now been incorporated. At present the EMPD remains associated with, but physically independent of the EPD. It is also subject to only periodic updates. In future we expect both the EPD and EMPD to become fully incorporated into the global Neotoma Palaeoecological Database, which will provide seamless integration of the fossil and modern data, whilst also allowing continual updates using Neotoma data management tools.

**8 Author contribution**

BASD wrote the manuscript with input from all of the authors. BASD, MC and PS designed and implemented the database and data viewer. BASD, MC, PS, MZ, WF, LNP, AM, and VC all helped with data processing.

**9 Competing interests**

The authors declare that they have no conflict of interest.

**10 Acknowledgements**

The EMPD includes data obtained from the Neotoma Palaeoecology Database and the European Pollen Database, and the work of the data contributors and the scientific community supporting these databases is gratefully acknowledged. Funding support for this project came from a Swiss National Science Foundation grant 200021_169598 'HORNET project' to BASD, with further support from the University of Lausanne.

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



**Tables**

| EMPD Metadata Fields | |
|---|---|
| **Sample Name** | Original authors sample name |
| **Sigle** | EMPD unique sample identifier |
| **Site Name** | Original authors site name |
| **Country** | Country where the site is located |
| **Longitude** | Longitude in decimal degrees |
| **Latitude** | Latitude in decimal degrees |
| **Elevation** | Elevation in meters asl |
| **Location Reliability** | Estimate of the accuracy of the geolocation information |
| **Location Notes** | Notes about the site location |
| **Area Of Site** | Site size in hectares |
| **Sample Context** | Physical environment of the site |
| **Site Description** | Notes about the physical context of the site |
| **Vegetation Description** | Notes about the surrounding vegetation |
| **Sample Type** | The type of material or sediment sampled |
| **Sample Method** | The method used obtain the pollen sample |
| **AgeBP** | The age of the sample BP |
| **Age Uncertainty** | The age uncertainty associated with the sample |
| **Notes** | General notes concerning the sample/site |
| **Publication 1-4** | Any publications associated with the sample |
| **Worker Role** | The name of the responsible person or analyst |
| **Worker Details** | Address and contact details for this person |

**Table 1**: List of metadata fields used in the EMPD




| Sample(s) | Country | Contributor(s) | Publication(s) |
|---|---|---|---|
| 3 | Belarus | Binney, H. | Binney et al., 2016, 2017 |
| 41 | Bulgaria | Atanassova, J., Lazarova, M., Tonkov, S. | Atanassova, 2007; Lazarova et al., 2006 |
| 33 | China, Peoples Republic Of | Binney, H. | Binney et al., 2016, 2017 |
| 56 | Cyprus | Fall, P. | Fall, 2012 |
| 47 | Czech Republic | Svobodova Svitavska, H. | Helena, 2004; Pardoe et al., 2010; Svobodová, 1989, 1997, 2002; Svobodová et al., 2001 |
| 1 | Finland | Stivrins, N. | Stivrins et al., 2017b |
| 4 | France | Leroy, S. | |
| 4 | Georgia | Binney, H. | Binney et al., 2016, 2017 |
| 85 | Germany | Giesecke, T., Matthias, I., Mrotzek, A., Rösch, M., Stojakowits, P. | Lechterbeck, 2001; Matthias et al., 2012, 2015; Mrotzek et al., 2017; Rösch et al., 2017; Rösch, 2009, 2012, 2013, 2018; Rösch and Lechterbeck, 2016; Rösch and Tserendorj, 2011a, 2011b; Rösch and Wick, 2019; Stojakowits, 2015 |
| 76 | Greece | Jahns, S., López Sáez, J., Mavridou, A., Panajiotidis, S., Papadopoulou, M., Tsakiridou, M. | Glais et al., 2016; Jahns, 1992; Pardoe et al., 2010 |
| 64 | Greenland | Edwards, K., Schofield, J. | Schofield et al., 2007 |
| 4 | Iceland | Hallsdottir, M., Stivrins, N. | |
| 16 | India | Demske, D., Tarasov, P. | Leipe et al., 2014 |
| 64 | Iran, Islamic Republic Of | Djamali, M., Leroy, S., Ramezani, E. | Djamali et al., 2009; Haghani et al., 2016; Leroy et al., 2011, 2018; Ramezani et al., 2013 |
| 243 | Italy | Accorsi, C., Badino, F., Champvillair, E., Clò, E., Colombaroli, D., Di Rita, F., Finsinger, W., Florenzano, A., Furlanetto, G., Greggio, B., Joannin, S., Leroy, S., Lotter, A., Magri, D., Mercuri, A., Montanari, C., Rattighieri, E., Ravazzi, C., Suanno, C., Tinner, W., Valsecchi, V. | Abbate, 1981; Finsinger et al., 2007, 2010; Florenzano et al., 2017; Florenzano and Mercuri, 2018; Furlanetto et al., 2019; Guido et al., 1992; Joannin et al., 2012; Margaritelli et al., 2016; Mercuri et al. 2012; Montali et al. 2006; Montanari and Guido, 1994; Rattighieri et al., 2010, 2012; Di Rita et al., 2011, 2018a, 2018b; Di Rita and Magri, 2009 |
| 84 | Japan | Kitaba, I., Leipe, C., Nakagawa, T., Watanabe, M. | Leipe et al., 2018 |
| 5 | Kazakhstan | Duryagina, N., Naidina, O., Nepomilueva, N. | Naidina and Richards, 2018; Nepomilueva and Duryagin, 1990 |





| 43 | Kyrgyzstan | Beer, R., Morales-Molino, C., Tinner, W. | Beer et al., 2007 |
|---|---|---|---|
| 10 | Latvia | Stivrins, N. | Feurdean et al., 2017; Grudzinska et al., 2017; Stivrins et al., 2014, 2015a, 2015b, 2016b, 2016a, 2017a; Veski et al., 2012 |
| 120 | Morocco | Alba-Sánchez, F., Fletcher, W., Sabariego Ruiz, S. | Bell and Fletcher, 2016 |
| 231 | Norway | Hjelle, K., Pardoe, H. | Caseldine and Pardoe, 1994; Hjelle et al., 2015; Hjelle and Sugita, 2012; Mehl and Hjelle, 2016; Pardoe, 1992, 2001, 2006, 2014 |
| 115 | Poland | Gałka, M., Karpińska-Kołaczek, M., Kołaczek, P., Latałowa, M., Milecka, K., Pędziszewska, A., Tobolski, K., Zimny, M., Święta-Musznicka, J. | Gałka et al., 2014, 2017; Milecka et al., 2017; Pardoe et al., 2010; Pędziszewska, 2008; Pędziszewska et al., 2015; Pędziszewska and Latałowa, 2016; Pidek et al., 2010 |
| 12 | Portugal | Fletcher, W. | Fletcher, 2005 |
| 17 | Romania | Feurdean, A., Grindean, R., Tantau, I. | Fărcaş and Tanţău, 2012; Feurdean et al., 2009, 2013, 2015; Feurdean and Willis, 2008a, 2008b; Grindean et al., 2014, 2015; Tanţău et al., 2014a, 2014b, 2009, 2011 |
| 1883 | Russian Federation | Antipina, T., Aseev, N., Belyanina, N., Binney, H., Blyakharchuk, T., Borisova, O., Bukreeva, G., Duryagin, D., Duryagina, N., Dyuzhova (Krasnorutskaya), K., Erokhin, N., Feurdean, A., Galimov, A., Golubeva, Y., Gvozdeva, I., Herzschuh, U., Ivanov, S., Karaulova, L., Khaymusova, N., Khizhnyak, N., Kremenetsky, N., Lapteva, E., Lopatina, D., Makovsky, N., Makovsky, V., Marchenko-Vagapova, T., Marieva, N., Matishov, G., Mikishin, Y., Müller, S., Naidina, O., Nepomilueva, N., Niemeyer, B., Nikiforova, L., Nosevich, E., Nosova, M., Novenko, E., Panova, N., Panova, N., Petrenko, T., Pisareva, V., Pisareva, N., Plotnikova, N., Ryabogina, N., Salonen, J., Sapelko, T., Semochkina, T., Seppä, H., Severova, E., Stivrins, N., Surova, | Antipina et al., 2014, 2016; Aseev, 1959; Binney et al., 2016, 2017; Blyakharchuk et al., 2007, 2019; Blyakharchuk and Chernova, 2013; Borisova et al., 2011; Bukreeva et al., 1986; Duguay et al., 2012; Hijmans et al., 2005; Ivanov and Ryabogina, 2004; Klemm et al., 2013, 2016; Kosintsev et al., 2010; Lapteva, 2009; Lapteva et al., 2013; Lapteva, 2013; Lapteva et al., 2017; Lapteva and Korona, 2012; Larin and Ryabogina, 2006; Lopatina and Zanina, 2016; Lychagina et al., 2013; Makovsky and Panova, 1978; Matishov et al., 2011; Matveev et al., 1997; Matveeva et al., 2003; Mikishin and Gvozdeva, 2009, 2012; Müller et al., 2010; Naidina and Richards, 2018; Nepomilueva and Duryagin, 1990; Niemeyer et al., 2017; Nikiforova, 1978; Novenko et al., 2011, 2014, 2017; Panova, 1981; Panova et al., 1996, 2010, 2008; Panova and Korotkovskaya, 1990; Panova and Makowski, 1979; Petrenko et al., 2009; Poshekhonova et al., 2008; Ryabogina and Orlova, 2002; Salonen et al., 2011, 2012; Sapelko and Nosevich, 2013; Shavnin et al., 2006; Stivrins et al., 2017b; Surova and Troitsky, 1971; Zakh, 1997 |





| | | N., Troitskiy, N., Vlasta Jankovska, N., Volkova, O., Yankovska, N., Zanina, O., Zelikson, E., Zhuykova, I. | |
|---|---|---|---|
| 134 | Spain | Alba-Sánchez, F., Anderson, R., García-Moreiras, I., Jiménez-Moreno, G., Leroy, S., López-Sáez, J.A., Mesa-Fernández, J., Morales-Molino, C., Muñoz Sobrino, C., Ramos-Román, M., Sabariego Ruiz, S. | Anderson et al., 2011; García-Moreiras et al., 2015; Jiménez-Moreno et al., 2013; Jiménez-Moreno and Anderson, 2012; Leroy, 1990; Mesa-Fernández et al., 2018; Morales-Molino et al., 2017a, 2018, 2011, 2013, 2017b; Morales-Molino and García-Antón, 2014; Muñoz Sobrino et al., 2014; Ramos-Román et al., 2016, 2018 |
| 4 | Sweden | Nielsen, A., Åkesson, C. | Åkesson et al., 2015; Ning et al., 2018 |
| 29 | Tunisia | Desprat, S., Gaceur, E. | Gaceur et al., 2017 |
| 31 | Turkey | Shumilovskikh, L. | |
| 2 | Ukraine | Binney, H., Borisova, O. | Binney et al., 2016, 2017 |
| 18 | United Kingdom | Bunting, M. | |

**Table 2**: List of data submitted to the EMPD2 by country




| Eurasian Modern Pollen Database (EMPD) | |
|---|---|
| Viewer: | https://empd2.github.io/?branch=master |
| Data Link: | https://epdweblog.org/european-modern-pollen-database/ |
| **European Pollen Database (EPD)** | |
| Viewer: | http://www.europeanpollendatabase.net/fpd-epd/bibli.do |
| Data Link: | https://epdweblog.org/epd_data/ |
| **Neotoma Paleoecology Database (NEOTOMA)** | |
| Viewer: | https://apps.neotomadb.org/explorer/ |
| Data Link: | https://www.neotomadb.org/data |
| **African Pollen Database (APD)** | |
| Viewer: | http://fpd.sedoo.fr/fpd/bibli.do |
| Data Link: | http://fpd.sedoo.fr/fpd/english.do |
| **Pangaea Data Archive (PANGAEA)** | |
| Viewer: | https://www.pangaea.de |
| Data Link: | https://www.pangaea.de |

**Table 3**: Web addresses for pollen databases mentioned in the text




**Figures**

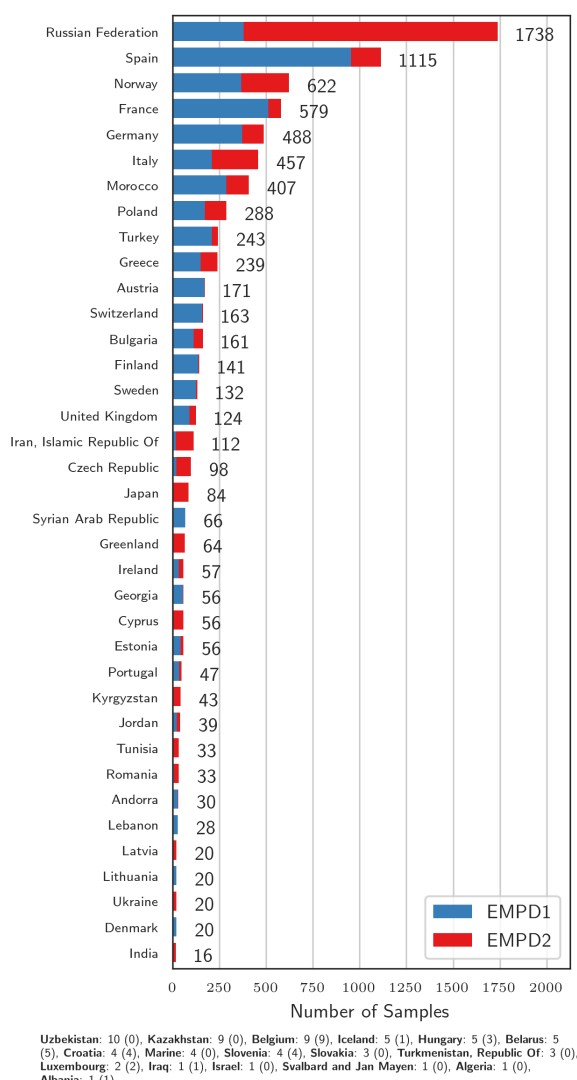

Uzbekistan: 10 (0), **Kazakhstan**: 9 (0), **Belgium**: 9 (9), **Iceland**: 5 (1), **Hungary**: 5 (3), **Belarus**: 5 (5), **Croatia**: 4 (4), **Marine**: 4 (0), **Slovenia**: 4 (4), **Slovakia**: 3 (0), **Turkmenistan, Republic Of**: 3 (0), **Luxembourg**: 2 (2), **Iraq**: 1 (1), **Israel**: 1 (0), **Svalbard and Jan Mayen**: 1 (0), **Algeria**: 1 (0), **Albania**: 1 (1)


**Figure 1**: A comparison of the number of samples in EMPD version 1 & 2, by country. Countries with only small numbers of samples are listed at the bottom, values in brackets indicate new samples in EMPD2.






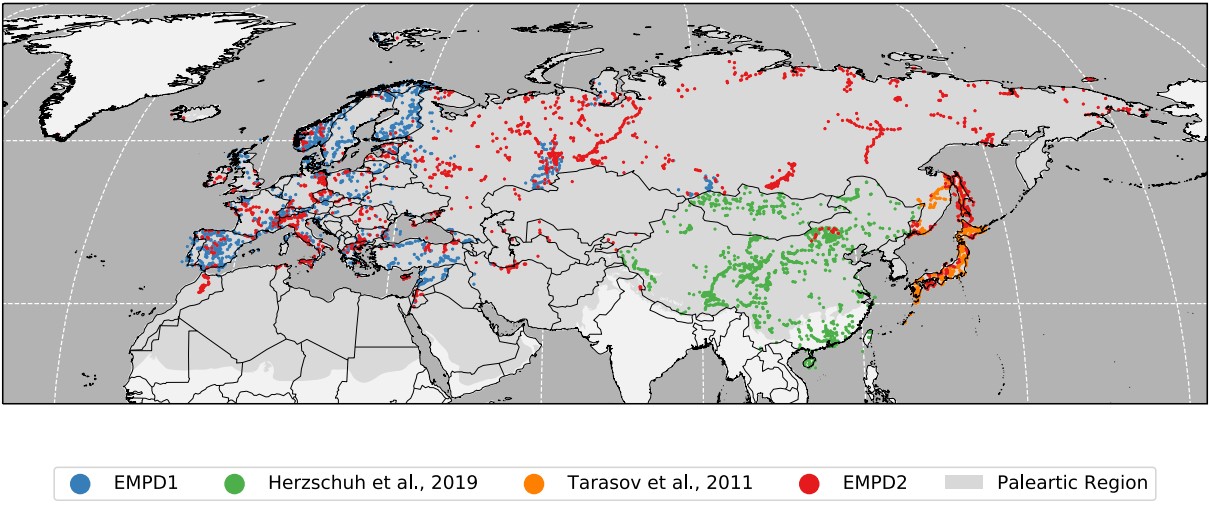

**Figure 2**: Map of samples included in the EMPD version 1 and 2, and two other datasets (see text)



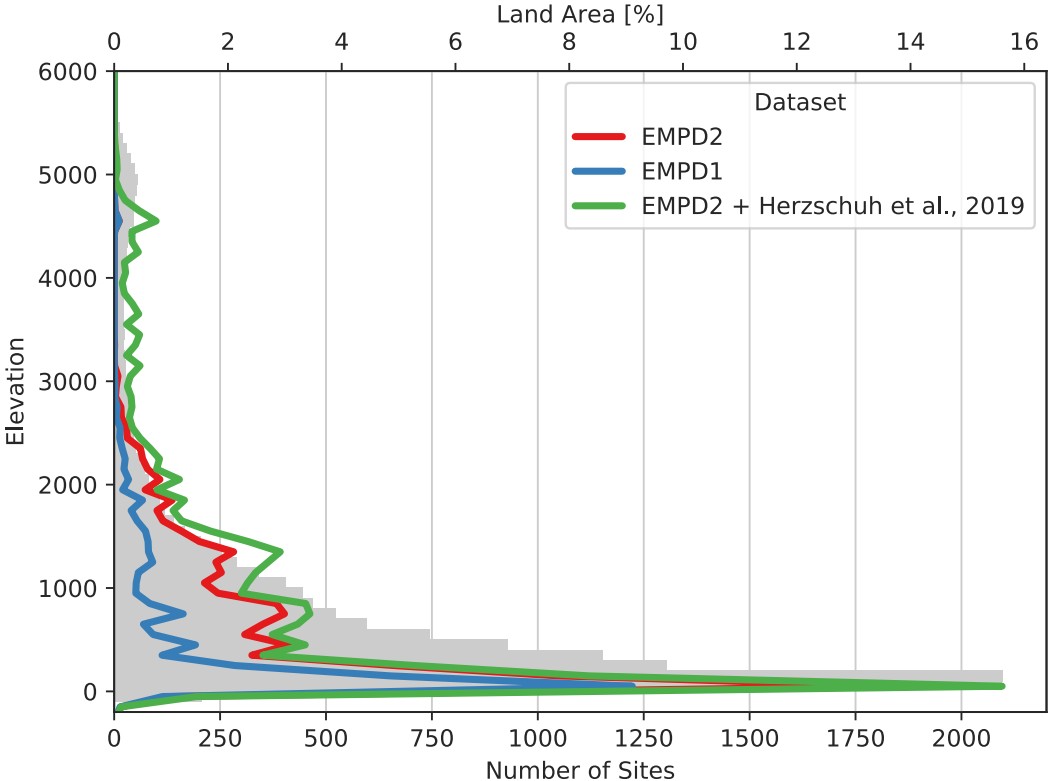

**Figure 3**: Distribution of samples by altitude for the Palearctic region (compared to land area at each altitude)





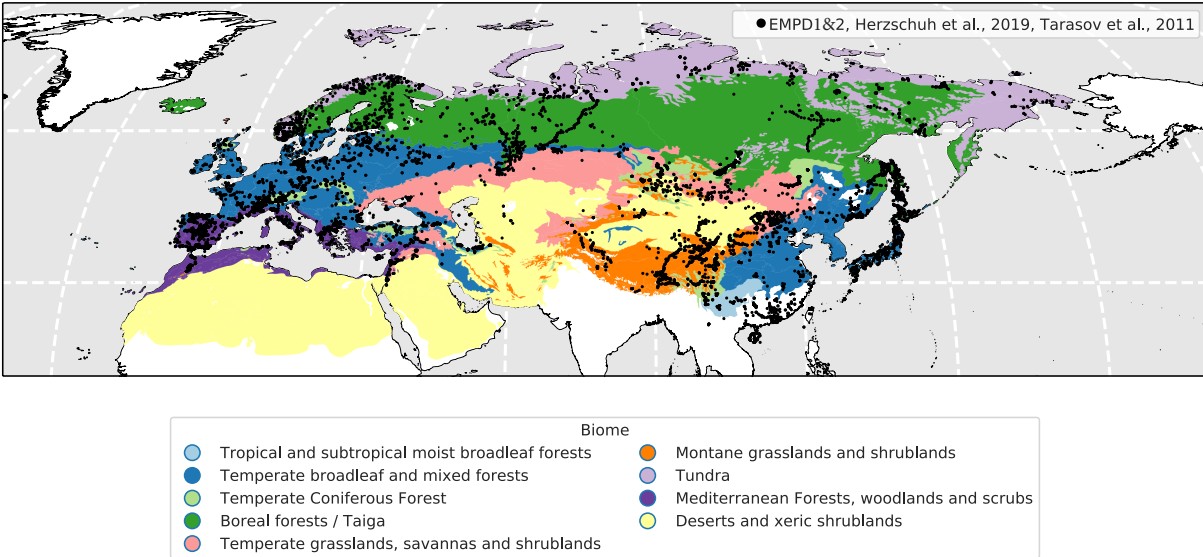

4a)

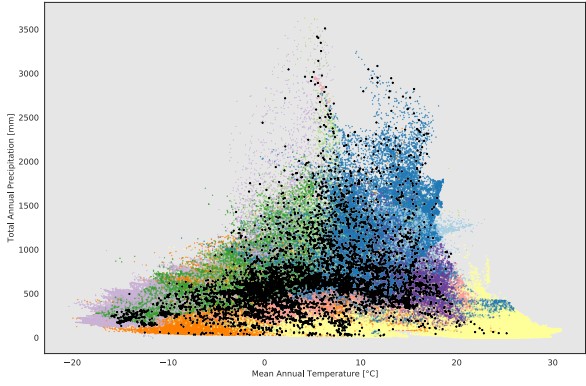

4b)

**Figure 4a**: Biome map and sample locations, **4b**: Biomes and samples in climate space. Biome data from Olson et al. (2001)






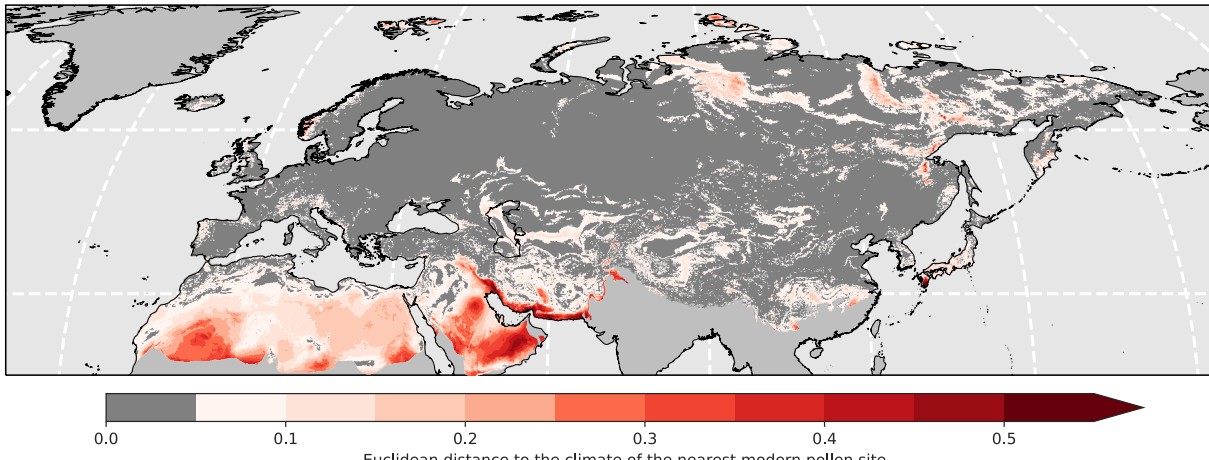

5a)

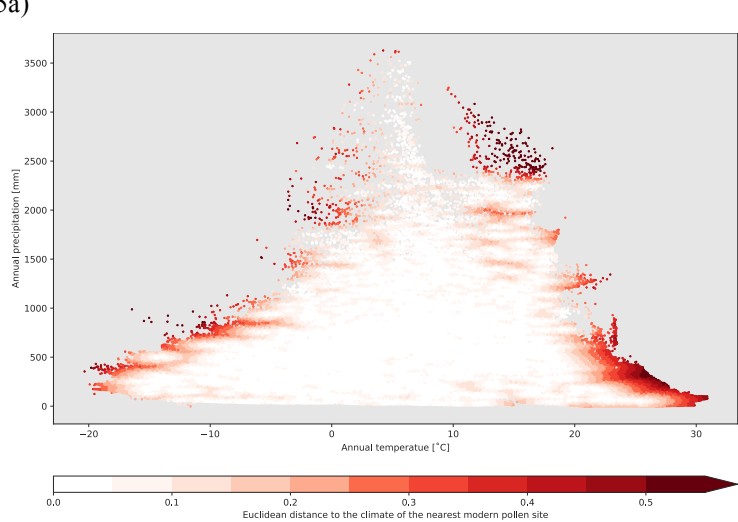

5b)


**Figure 5a**: The euclidian distance between the climate of each modern pollen sample location (as shown in Fig. 2) and the climate of the entire Palarctic region; **5b:** The same as 5a but shown in climate space. Note that for clarity the values <0.05 are shown by dark grey in 5a, but white in 5b. The darker the brown shading, the less well that climate is represented amongst the samples.






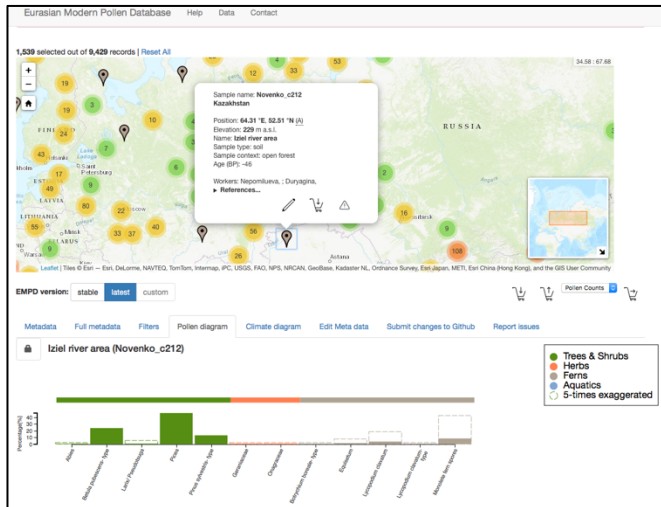

**Figure 6**: Screen grab of the EMPD online data viewer (available at https://empd2.github.io)