# Peer review of "The Eurasian Modern Pollen Database (EMPD), Version 2"

_Earth System Science Data, 2020_

## Referee Comment (RC1) · Anonymous Referee #1 · 17 Mar 2020

This study provides an updated EMPD dataset (i.e., EMPD2), which contains more modern pollen samples and covers a larger spatial extent than EMPD1. Such dataset is very useful to reconstruct past vegetation and climate. The new map-based online data viewer is very interesting and can advance data sharing and quality checking. The manuscript falls well within the scope of the journal and provides a contribution to ESSD. However, its presentation and structure should be greatly improved to meet ESSD's requirement. Therefore, I recommend it could be accepted after a major revision or resubmission.

Two major concerns 1) As we are informed by the submission guidelines, ESSD is focused on how the data were processed, such as quality controlling, new technique used, and etc.. However, the authors ignored the issue. So I suggest the authors

should strengthen the section 2.2 with more details (e.g. flowchart) on the data processing. Additionally, I think it will be better to combine 2.3 with 2.2. 2) The results and discussion should be deepened and extended since the contents has not acknowledged readers some new knowledge.

Specific comments: 1. P4L136-138: The sentence "although it is in fact more accurately described ..." is complicated and not straightforward, please reword it. 2. P4L144-145: Please restructure the sentence "... interpret past change in land cover, land use and human impact ..." 3. P5L177-179: "and compatibility with the EPD" -> which 4. P5L179-181: There is a lot of repetition, e.g., "it was collected", please rewrite. 5. P6L231: add "loose" before "definition" and delete "was more loosely defined". 6. P9L326: not clear to me how the Euclidean distance is calculated? Does "the climate of each of the pollen samples" refer to temperature or precipitation value? mean monthly or seasonal or annual value, which one did you choose? Did you calculate the Euclidean distance at grid scale with resolution of 30 second (Fig.5a)? Please improve the description and consider illustrate it in the method section. 7. P10L336: please reword "This will make possible more accurate reconstructions". 8. The labels of X and Y in Figs. 4b and 5b should be enlarged.

---

## Referee Comment (RC2) · Anonymous Referee #2 · 21 Mar 2020

The authors present an update of a database of modern pollen samples from Eurasia. These databases have a wide range of applications, and having standardized, robust and easily available data is a great benefit to the community. The update is substantial, not just in terms of the number of records, but in the increase in geographical range and climate space covered by the samples, and a new publication for this update is well justified.

The paper is well written, with just a few minor things to check (given below). Ordinarily I would have liked to see more detail on the application, but this is well described in the paper for the first database version. The new work on demonstrating the coverage in various parameter spaces (geographical, land cover, climate) is very neat, and helps demonstrates the strengths and limitations of the data, and may hopefully guide future

data collection efforts. Overall, I recommend this for publication.

Minor comments:

Line 177: '6' should be '7'

Line 263-264: How has the taxon harmonization been done? Are the original taxonomic names retained in case of problems? Being able to match the data to fossil records is vital, so a little more detail would be useful here. Not all of the taxa are harmonized – but are the remaining one not found in the fossil records?

Line 301: This is a minor issue, but the traditional definition of a hypsometric curve is a cumulative frequency curve.

Figures 1,2,3: The color choices here may be a problem for color-blind readers. Can the authors use different line types or symbols?

―――――――――――――――――

---

## Author Comment (AC1) · 13 May 2020

We would like to thank the reviewers for taking the time to read and comment on our manuscript. The comments from the reviewers and our response and action to each of these comments is provided below.

A clearer version of these comments is provided as a supplementary file. In this supplementary version the reviewers comments are shown in bold italic and any new or amended text is highlighted in yellow.

Response to reviewer 1

Two major concerns 1) As we are informed by the submission guidelines, ESSD is focused on how the data were processed, such as quality controlling, new technique

used, and etc.. However, the authors ignored the issue. So I suggest the authors should strengthen the section 2.2 with more details (e.g. flowchart) on the data processing.

Response: The issue of data processing is not ignored, it is included in the section 2.2 "Data processing", where clear reference is made to an earlier paper where more detail on this topic is provided; lines 254-255 "files had to be processed and a variety of quality control checks made (see Davis et al., 2013) before entry into the database." In section 2.2 of the manuscript we focus on those elements of the processing which are different from this earlier paper.

We can however provide more information and a flowchart to make the steps taken clearer.

Action: We now include a flowchart (figure 1) to show the steps taken in data processing, as well as the following text to section 2.2:

Figure 1 shows the steps taken in processing and quality controlling the data. On receipt from the contributor, the data was entered into one of two standardized file formats according to whether it was pollen data or the associated metadata. Each of the two different types of data was then subject to a series of quality control checks to make sure it did not contain errors and that it conformed to data protocols. For instance, values in numerical fields in the metadata (shown in table 1) had to fall within realistic boundaries expected for that field, such as for latitude, longitude and altitude. Also, that controlled fields based on selection from a list of acceptable classes did not contain assignment errors, such as country name. Any missing entries were referred back to the contributor for completion, or else were completed from the original publication or other information source where available.

Once the pollen data and metadata entry tables had been manually completed and checked, these were then uploaded into a Postgres database where a second series of automated quality control procedures were undertaken. These automated checks repeated many of the earlier manual checks, including ensuring that all open and closed fields were correctly completed, and that the taxa names conformed to the database standardised taxa names (the 'p_vars' table). In addition, it was also necessary to manually standardize worker names, address details, and data references across different datasets submitted to the database.

After the data had passed these database checks, each contributor was then asked to look again at their data as it was now stored in the database. Contributors were able to do this using the online data viewer, which provided an intuitive interface to the database that could be navigated without any prior experience of database systems. Locations for each site/sample could be checked using the viewer map interface, pollen data could be checked using a graphical (histogram) display, and metadata checked using a table view of all of the metadata fields. Any issues highlighted by the contributors were then corrected in the database. It was only after completing these final contributor checks was the EMPD2 database deemed suitable for public release.

Additionally, I think it will be better to combine 2.3 with 2.2.

Response: OK

Action: Sections 2.2 and 2.3 have been combined

2) The results and discussion should be deepened and extended since the contents has not acknowledged readers some new knowledge.

Response: It would help if the reviewer could provide some additional guidance to help understand their statement. This is a data descriptor paper submitted to a journal for the publication of articles on original research data (sets). In the results and discussion, we describe the data set and how representative it is of the geographical vegetation and climatic space from which it has been collected. Within this we highlight the strengths and weaknesses of the dataset, and how it might be improved in the future. If the reviewer can think of something of particular importance that we have left out then we

would be very happy to include this, although we note that reviewer 2 did not have any issue with the results and discussion.

Action: None

Specific comments: 1. P4L136-138: The sentence "although it is in fact more accurately described : : :" is complicated and not straightforward, please reword it.

Response: Ok.

Action: The sentence "We use the term fossil pollen here as it is used in the Quaternary sciences, although it is in fact more accurately described as sub-fossil having usually undergone only limited (if any) mineralization, and including many spores as well as pollen from flowering plants." has been changed to "We use the term 'fossil pollen' here as it is commonly used in the Quaternary sciences. The fossils in this sense can more accurately be described as sub-fossil since they have usually only undergone limited (if any) post-deposition mineralization, while pollen is taken to include many spores as well as the pollen from flowering plants."

2. P4L144-145: Please restructure the sentence ": : : interpret past change in land cover, land use and human impact : : :"

Response: Ok.

Action: The sentence "Modern pollen samples have been used to interpret past change in land cover, land use and human impact, the impact on vegetation of past edaphic change, the effects of fire, pests and disease, as well as hydroseral change." Has been changed to.. "Modern pollen samples have been used to interpret many different environmental processes, such as past changes in land cover, land use and human impact; the impact on vegetation of past edaphic and hydroseral changes; and the effects of past changes in fire, pests and disease on vegetation."

3. P5L177-179: "and compatibility with the EPD" - > which

Response: Ok.

Action: The sentence.. "Importantly, all of these aspects limit their compatibility with the EPD, and compatibility with the EPD is one of the primary objectives of the EMPD." Has been changed to "Importantly, all of these aspects limit their compatibility with the EPD, where compatibility with the EPD is one of the primary objectives of the EMPD"

4. P5L179-181: There is a lot of repetition, e.g., "it was collected", please rewrite.

Response: The use of repetition is a deliberate rhetorical device common in the English language. For the sentence in the text where the word 'collected' is repeated at the end of each clause, this is called Epistrophe. For example, "that government of the people, by the people, for the people" (Abraham Lincoln). The sentence in question in the text does not appear to have been problematic to reviewer 2, who sounds like a native speaker: "The EMPD also contains comprehensive and standardized metadata about the pollen sample location, the landscape and vegetation environment from which it was collected, the way it was collected, the year that it was collected, as well as who collected and analyzed the sample and where it was published."

Action: None.

5. P6L231: add "loose" before "definition" and delete "was more loosely defined".

Response: The referee suggests changing the sentence from "Another problem with this heritage data apart from the limited metadata, is that the definition of a 'modern sample' was more loosely defined in these early projects, being defined in both PAIN and Biome6000 as anything younger than 500 BP." to "Another problem with this heritage data apart from the limited metadata, is that the loose definition of a 'modern sample' in these early projects, being defined in both PAIN and Biome6000 as anything younger than 500 BP." However, this doesn't make grammatical sense unless you also remove the 'that' before 'loose'.

Action: The sentence has been changed to "Another problem with this heritage data
apart from the limited metadata is the loose definition of a 'modern sample' in these early projects, being defined in both PAIN and Biome6000 as anything younger than 500 BP."

6. P9L326: not clear to me how the Euclidean distance is calculated? Does "the climate of each of the pollen samples" refer to temperature or precipitation value? mean monthly or seasonal or annual value, which one did you choose? Did you calculate the Euclidean distance at grid scale with resolution of 30 second (Fig.5a)? Please improve the description and consider illustrate it in the method section.

Response: We have improved the description in line with the referees comments.

Action: The following text has been added to the discussion section to describe more clearly how Euclidean distance was calculated: "This was done using mean annual temperature and precipitation from the WorldClim2 modern climatology (Fick and Hijmans, 2017), normalized to make the different scales comparable. The climate of each pollen site was assigned according to the nearest grid point within the 30 second (approximately 1km2) resolution of the WorldClim2 grid, whilst the climate of the region was taken from the grid itself."

The following text was also added to the Figure 5 caption: "The climate of each pollen site was assigned according to the nearest grid point within the 30 second (approximately 1km2) resolution of the WorldClim2 grid, whilst the climate of the region was taken from the grid itself."

7. P10L336: please reword "This will make possible more accurate reconstructions".

Response: Ok.

Action: The following text has been changed from: "This will make possible more accurate reconstructions.." to: "This will make it possible to create more accurate reconstructions.."

8. The labels of X and Y in Figs. 4b and 5b should be enlarged.

[Figure]

Response: Ok.

Action: These labels have been enlarged.

Response to reviewer 2

The authors present an update of a database of modern pollen samples from Eurasia. These databases have a wide range of applications, and having standardized, robust and easily available data is a great benefit to the community. The update is substantial, not just in terms of the number of records, but in the increase in geographical range and climate space covered by the samples, and a new publication for this update is well justified.

The paper is well written, with just a few minor things to check (given below). Ordinarily I would have liked to see more detail on the application, but this is well described in the paper for the first database version. The new work on demonstrating the coverage in various parameter spaces (geographical, land cover, climate) is very neat, and helps demonstrates the strengths and limitations of the data, and may hopefully guide future data collection efforts. Overall, I recommend this for publication.

Minor comments:

Line 177: '6' should be '7'

Response: Ok.

Action: 6 has been changed to 7

Line 263-264: How has the taxon harmonization been done? Are the original taxonomic names retained in case of problems? Being able to match the data to fossil records is vital, so a little more detail would be useful here. Not all of the taxa are harmonized – but are the remaining one not found in the fossil records?

Response: Ok.

Action: We have added the following paragraph to clarify the taxa harmonization process in line with the referees comments: "The accepted names for the fossil data in the EPD or Neotoma should be directly compatible with the accepted names in the EMPD, but some caution needs to be applied in integrating the two datasets since the EMPD contains additional accepted names that do not occur in the EPD or Neotoma. Where possible the EMPD assignment of accepted names respects the taxonomic resolution of the EPD and Neotoma accepted names. This means that where a new original taxa name is submitted to the EMPD that does not already occur in the existing databases, it is assigned the EPD or Neotoma accepted name according to the existing taxonomic hierarchy. For example, if the new submitted original taxa name is a new species that does not occur in the EPD or Neotoma, and there is an existing accepted name at genus level, then the new species name is assigned the accepted name at the genus level. The assignment of accepted names is complicated because it requires an appreciation of differences in pollen morphology and of the reliability of identification, which can vary given the differences in skill and experience of the different analysts who contribute to the database. In addition, there are also important geographical considerations to take into account. For instance, the EMPD conforms to the EPD accepted names but these are heavily European orientated, while the EMPD has much more data from regions such as eastern Asia where some of the accepted names are not strictly appropriate. However, in all cases we have retained in the EMPD all of the original taxa names as they were submitted by the original contributor after cleaning for typographical errors."

Line 301: This is a minor issue, but the traditional definition of a hypsometric curve is a cumulative frequency curve.

Response: A hypsometric curve is a type of cumulative frequency curve based on elevation. The word 'hypsometric curve' is an accepted scientific term in geomorphology (eg. Lagrula 1997 https://doi.org/10.1007/3-540-31060-6_183).

Action: The text has been changed from: "plotting the distribution of samples by altitude on a hypsometric curve" to "plotting the distribution of samples by altitude on a hypsometric (or cumulative frequency) curve"

Figures 1,2,3: The color choices here may be a problem for color-blind readers. Can the authors use different line types or symbols?

Response: I am strongly red-green colour blind. The colours used in the figures have been specifically chosen to appear clear to people with red-green colour blindness. Figures 1 and 2 use only two and three colours respectively and these have been chosen to have contrasting tones as well as colours, which make them easier to differentiate for people with colour deficiency. Figure 2 uses four colours. The reviewer suggests using symbols, and we did originally try using symbols but we quickly realized that the symbols have to be quite large to be clearly differentiated from each other. This then creates a different problem since using large symbols makes it more difficult to see the location of the individual sites, while at the same time overlapping symbols obscure the symbol shape itself.

The other additional reason for favouring the use of colour in all of the figures is that it provides continuity between the figures, since the same colours are used to identify the same data source in all of the figures, irrespective of whether these use dots or lines.

Action: None.

Please also note the supplement to this comment:
https://www.earth-syst-sci-data-discuss.net/essd-2020-14/essd-2020-14-AC1-supplement.pdf
* * *
[Figure]

Figure 1. A flow diagram showing the data processing and quality control steps taken in constructing the EMPD2 database

**Fig. 1.** Figure 1